# Comparing Different Chemometric Approaches to Detect Adulteration of Cold-Pressed Flaxseed Oil with Refined Rapeseed Oil Using Differential Scanning Calorimetry

**DOI:** 10.3390/foods12183352

**Published:** 2023-09-07

**Authors:** Mahbuba Islam, Anna Kaczmarek, Magdalena Montowska, Jolanta Tomaszewska-Gras

**Affiliations:** 1Department of Food Quality and Safety Management, Poznań University of Life Sciences, ul. Wojska Polskiego 31/33, 60-624 Poznań, Poland; mahbuba.islam@up.poznan.pl (M.I.); anna.kaczmarek@up.poznan.pl (A.K.); 2Department of Meat Technology, Poznan University of Life Sciences, ul. Wojska Polskiego 31/33, 60-624 Poznań, Poland; magdalena.montowska@up.poznan.pl

**Keywords:** DSC melting profile, multivariate analysis, oils authenticity, plant oils, multiple linear regression, classification model, artificial neural networks (ANN), orthogonal partial least squares discriminant analysis (OPLS-DA), MARS, SVM

## Abstract

Flaxseed oil is one of the best sources of *n*-3 fatty acids, thus its adulteration with refined oils can lead to a reduction in its nutritional value and overall quality. The purpose of this study was to compare different chemometric models to detect adulteration of flaxseed oil with refined rapeseed oil (RP) using differential scanning calorimetry (DSC). Based on the melting phase transition curve, parameters such as peak temperature (T), peak height (h), and percentage of area (P) were determined for pure and adulterated flaxseed oils with an RP concentration of 5, 10, 20, 30, and 50% (*w*/*w*). Significant linear correlations (*p* ≤ 0.05) between the RP concentration and all DSC parameters were observed, except for parameter h1 for the first peak. In order to assess the usefulness of the DSC technique for detecting adulterations, three chemometric approaches were compared: (1) classification models (linear discriminant analysis—LDA, adaptive regression splines—MARS, support vector machine—SVM, and artificial neural networks—ANNs); (2) regression models (multiple linear regression—MLR, MARS, SVM, ANNs, and PLS); and (3) a combined model of orthogonal partial least squares discriminant analysis (OPLS-DA). With the LDA model, the highest accuracy of 99.5% in classifying the samples, followed by ANN > SVM > MARS, was achieved. Among the regression models, the ANN model showed the highest correlation between observed and predicted values (R = 0.996), while other models showed goodness of fit as following MARS > SVM > MLR. Comparing OPLS-DA and PLS methods, higher values of R^2^X(cum) = 0.986 and Q^2^ = 0.973 were observed with the PLS model than OPLS-DA. This study demonstrates the usefulness of the DSC technique and importance of an appropriate chemometric model for predicting the adulteration of cold-pressed flaxseed oil with refined rapeseed oil.

## 1. Introduction

Recent studies have provided convincing evidence that combining chemometric methods with analytical measurements can produce remarkable results in assessing food quality, in particular its authenticity. This novel approach provides for a more thorough and accurate evaluation of foodstuffs, allowing for the detection of any adulteration or mislabeling [1,2,3]. These findings highlight the need to combine data and use multivariate statistical analysis tools to verify the authenticity and quality in the food supply chain. Adulteration of high-priced edible oils continues to be a major concern for both the edible oil industry and consumer health, despite the fact that experts recognized the problem centuries ago [4]. This deceptive technique is primarily motivated by individuals seeking to increase their revenues by boosting the volume of the product [5] by taking advantage of the lack of effective quality assessment tools for unreliable food products, as suggested by the Food and Agricultural Organization (FAO) [6]. Food adulteration occurs frequently in many wealthy countries around the world, despite researchers’ attention to the fraudulent phenomenon [7]. Addressing food fraud poses a significant financial burden, with estimates suggesting that the global food industry incurs an annual cost of approximately EUR 30 billion. This substantial expense highlights the economic impact of combating fraudulent practices in the food sector [8].

Several approaches to investigating and detecting adulterations in food products have been proposed by various food scientists. Researchers have shown the successful application of combining analytical techniques with linear and non-linear chemometric tools [9] to build classification and regression models for oil samples, which clearly demonstrates the importance of different uses of chemometric techniques, i.e., linear discriminant analysis (LDA) [10], multiple linear regression (MLR) [2], multivariate adaptive regression splines (MARS) [11], support vector machine (SVM) [12], artificial neural networks (ANNs) [13], principle component analysis (PCA) [14], orthogonal partial least squares discriminant analysis (OPLS-DA) [15], and partial least squares regression (PLS) [16]. For instance, to detect adulteration in extra virgin olive oils, UV-IMS (ultraviolet ion mobility spectrometry) combined with chemometric analyses like PCA and LDA [17], near-infrared spectroscopy with chemometric techniques [18], and DSC combined with SVM [19] were used. The detection of adulteration in flaxseed oil has also been reported by other authors using different analytical methods coupled with a statistical approach, e.g., mid-infrared spectroscopy (MIR) associated with the chemometric technique of PLS [20], low-field nuclear magnetic resonance relaxation fingerprints [21], gas chromatography–mass spectrometry (GC-MS) coupled with PCA and recursive support vector machine (R-SVM) [22], HPLC-ELSD profiling of triacylglycerols and chemometrics [23], dielectric spectroscopy with PCA and LDA analysis [10], and Fourier transform infrared spectroscopy (FTIR) and MLR [24]. Most of these studies emphasized the importance of using multivariate methods to efficiently detect the adulteration. Different studies were also conducted to show the applicability of using the DSC technique for the adulteration assessment of different fats and oils, which are comparatively expensive and acclaimed as being nutritious, e.g., olive oils and other vegetable oils [25,26,27,28,29,30] and animal fats [31,32,33,34]. In pursuit of the idea of gap analysis, DSC stands out as an analytical method with the ability to detect changes associated with changes in the composition of triacylglycerols, which makes it possible to use it as a “at-a-glance” method for oil authentication. This method measures the thermodynamic parameters of temperature and enthalpy of phase transition without the use of any chemicals, which is not possible in the case of liquid chromatography. Unlike other methods like FTIR [24], XRD [35], and NMR [14], the thermal behavior of the material can be studied under different conditions, e.g., scanning rate. The ability of the DSC technique has already been proven to provide quantitative thermal data in fields such as pharmaceuticals [36], polymers [37], and food science [38].

Derived from the flax plant (*Linum usitatissimum* L.), flaxseed is a seed that is widely grown in countries like Canada, America, China, and India [39]. The oil obtained from cold-pressing flaxseeds, known as flaxseed oil, is highly regarded for its exceptional content of α-linolenic acid (ALA) [40,41], an essential fatty acid that can be converted into beneficial compounds like eicosapentaenoic acid (EPA) and docosahexaenoic acid (DHA) in the human body [42]. Additionally, flaxseed contains abundant phenolic compounds such as lignans, ferulic acid, and p-coumaric acid, as well as mucilage. These bioactive components have demonstrated positive effects on intestinal function [43]. Flaxseed oil has been found to offer numerous health benefits, particularly for the cardiovascular and skeletal systems. It has also shown positive effects in inflammatory conditions like rheumatoid arthritis, psoriasis, ulcerative colitis, and colon tumor [44,45]. An authenticity analysis is crucial for cold-pressed oils like flaxseed oil, as they are commonly targeted by fraudulent practices, such as adulteration with lower-quality oils or blending with cheaper oils. Refined rapeseed oil can be used as an adulterant to flaxseed oil, since it is cheaper and widely used as cooking oil known for its neutral flavor, high smoke point, and longer shelf life. Adulteration of flaxseed oil with refined rapeseed oil compromises its authenticity and can lead to a reduction in its nutritional value and overall quality [46]. Therefore, the aim of the study was to study the feasibility of the DSC technique combined with chemometric methods for the detection of adulteration of cold-pressed flaxseed with refined rapeseed oils in different concentrations. The novelty of the study lies in evaluating the potential of DSC coupled with various chemometric methods, which were employed to create predictive classification and regression models for the detection of oil adulteration. The classification approach categorized the level of oil adulteration, while the regression approach treated the concentration of refined rapeseed oils as a continuous variable. A number of chemometric methods, i.e., LDA, MLR, MARS, SVM, ANNs, PCA, OPLS-DA, and PLS were used to classify, describe, and generate prediction models of the adulteration phenomena. The originality of the study is evidenced by the pioneering comparative analyses of different chemometric models, which show both the good results of the LDA model in identifying adulterated flaxseed oil samples and the regression model in which ANN excels at predicting adulterated concentrations. This systematic exploration of various chemometric techniques highlights the researchers’ innovative approach to increasing the accuracy of the detection and classification of adulterated oils.

## 2. Materials and Methods

### 2.1. Materials

Sample oil seeds for cold-pressed flaxseed oils were obtained from different Polish cultivars and then pressed mechanically. Seeds from four different cultivars were purchased, i.e., *Bukoz* (Polish Institute of Natural Fibers and Medicinal Plants in Poznan), *Dolguniec*, *Szafir* (SEMCO manufactory in Śmiłowo, the Hodowla Roślin Strzelce Sp. z o.o. in Strzelce), and one sample of an unidentified variety from the VitaCorn company in Poznan. The oils were obtained through screw-pressing the seeds while keeping temperature below 50 °C at the SEMCO manufactory (Śmiłowo, Poland). After the pressing process, the oils were left for 24 h for decantation and subsequently stored in brown glass bottles. Refined rapeseed oil was purchased from the market. Cold-pressed flaxseed oil samples were adulterated by adding refined rapeseed oils in varying concentrations (0, 5, 10, 20, 30, and 50% *w*/*w*). Prepared samples were analyzed in three replications.

### 2.2. Methods

#### 2.2.1. Melting Phase Transition Analysis via Differential Scanning Calorimetry (DSC)

Melting analysis of oil samples was carried out with modifications according to the method used for butterfat [3,47]. A Perkin Elmer differential scanning calorimeter DSC 8500 PerkinElmer (Waltham, MA, USA), equipped with an Intracooler II and running with Pyris software 11 (Perkin Elmer, Waltham, MA, USA), was used. Nitrogen (99.999% purity) was the purge gas. Samples of ca. 6–7 mg were weighed into aluminum pans of 20 µL (Perkin Elmer, No. 0219-0062, Waltham, MA, USA) and hermetically sealed. An empty, hermetically sealed aluminum pan was used as reference. Analysis started with cooling the oil sample at a scanning rate of 2 °C/min from a temperature of 30 °C to −65 °C, after which it was heated at scanning rates 5 °C/min from −65 °C to 30 °C. For each measurement at a given scanning rate, the calibration procedure was completed with the correct scanning rate. After the analysis, the DSC files were converted to the ASCII format and then were analyzed using Origin Pro software, version 2023 (OriginLab Corporation, Northampton, MA, USA). The Origin PeakFit module was used to project the DSC curves of all investigated samples. Different DSC parameters, i.e., peak temperature (T, °C), peak height (h, W/g), and percentage of area (P, %), were measured from the melting curves. Peak temperature (T) was determined on the temperature axis (X) by locating the maximum point of heat flow for each peak. Peak height (h) was determined at the heat flow maximum on the axis Y for each peak. The percentage of each peak area (P) was calculated as the ratio of the area of each peak to the total area of the melting phase transition curve.

#### 2.2.2. Data Analysis

The recorded data were subjected to statistical analysis using Statistica 13.3 software, developed by TIBCO Software Inc. in the USA. A significance level of α = 0.05 was chosen for the analysis. The outcomes were reported as the mean and standard deviation. The initial step in the statistical analysis involved assessing the assumptions of ANOVA, which included testing for variance homogeneity and checking data normality. If these assumptions were met, one-way analysis of variance (ANOVA) was performed, followed by the application of Tukey’s test to establish statistically homogeneous groups. To assess the impact of adding adulterants at varying percentages to cold-pressed flaxseed oils, linear regression analysis was employed, utilizing the least squares estimation method. This analysis considered the DSC parameters extracted from the melting curves. Classification and regression approaches were used to build predictive models for oils adulteration detection as it was proposed by authors for the food analysis purpose [9,48]. For all models, the predictors were DSC parameters, while the dependent variable was the level of oil adulteration. In the classification approach, the dependent variable (level of oil adulteration) was categorial (6 classes—one for each level of oil adulteration), while in the regression approach, the dependent variable was a continuous variable (concentration of refined rapeseed oils). Artificial neural networks (ANNs), support vector machine (SVM), and multivariate adaptive regression splines (MARS) were used to build classification as well as regression models. In addition, linear discriminant analysis (LDA) was used to build a classification model and multiple linear regression analysis (MLR) to build a regression model. The leave-one-out cross-validation was used. The goodness of fit of the regression models was estimated based on R (correlation coefficient), R^2^ (determination coefficient), adjusted R^2^ (modified version of R^2^ was adjusted for the number of predictors in the model), Akaike information criterion (AIC), Bayesian information criterion (BIC), and Root Mean Square Error (RMSE). The confusion matrix was used for selecting the best classification model. The confusion matrix represents the counts of predicted and true values. The score “TN” stands for True Negative, which shows the number of correctly classified negative examples. Similarly, “TP” stands for True Positive, which indicates the number of correctly classified positive examples. “FP” stands for False Positive, which is the number of actual negative examples classified as positive, and “FN” stands for False Negative, which is the number of actual positive examples classified as negative. Based on the confusion matrix, the performance parameters (Accuracy, Misclassification Rate, Precision, Sensitivity, Specificity, and F1-score) of the classification models were calculated. To perform PCA and OPLS-DA, the SIMCA software version 16.1 (Sartorius Stedim Data Analytics AB, Umea, Sweden) was utilized. Cross-validation was performed on both the PCA and OPLS-DA models, and the OPLS-DA models were validated using permutation testing. For these models, DSC parameters were considered as X variables, and Y variables consisted of different levels of concentrations added as adulterants to the flaxseed oils. The metrics (R^2^X, R^2^ and Q^2^) were analyzed from the models to collectively provide information about how well the OPLS-DA model fits the X data (DSC parameters).

## 3. Results

### 3.1. DSC Melting Profiles of Cold-Pressed Flaxseed Oil Adulterated with Refined Rapeseed Oil

In Figure 1, the DSC melting curves obtained for all cultivars of flaxseed oils (pure and adulterated) are presented. Figure 1a–e each demonstrate the melting curves for different cultivars of flaxseed oils (*Bukoz*, *Dolguniec*, *Szafir* (A, B), and unknown variety, respectively) with different concentrations of refined rapeseed oil (0, 5, 10, 20, 30, and 50% *w*/*w*). In this study, the thermal profile of flaxseed oils was examined using melting DSC curves to track the alterations resulting from the addition of adulterants (rapeseed oil). All the samples were first crystallized to −65 °C at a 2 °C/min cooling rate prior to the heating program. On the melting curve of pure flaxseed oil, three endothermic peaks were identified as a result of the melting of crystals and nuclei. The first shoulder peak was detected at around −36 °C; the second, as a major peak, occurred at around −30 °C; and the third peak appeared at a temperature of approx. −25 °C (Figure 1). Apart from the flaxseed oil, the melting curves of refined rapeseed oil, which gives two endothermic peaks, are also shown.

Between the DSC curve for flaxseed oil and refined rapeseed oil curves, the curves of adulterated flaxseed oil with rapeseed in different concentrations are shown. Comparing Figure 1a–e, it can be seen that all the varieties have shown similar changes of thermal transition profile upon the addition of refined rapeseed oil. With an increasing refined oil concentration, gradual changes in the formation of all peaks can be observed in Figure 1. Apparently, all three peaks’ positions were shifted to a higher temperature with the gradual addition of rapeseed oil. Consequently, the peak height also changed, because the main peak decreased, while the height of the side peaks (first and third) increased with the increase in the concentration of rapeseed oils in the mixture. Alterations in the thermodynamic characteristics of the target samples with the addition of adulterants have been reported in other studies as well. A similar approach to adulterating cold–pressed oils with refined and cheaper oils was taken by other authors studying adulteration. For instance, using mid-infrared spectroscopy (MIR), authors performed a quantitative analysis of soybean oil and sunflower oil as adulterants with concentrations from 3.5 to 30% (*w*/*w*) in extra virgin flaxseed oil [20]. Another study showed the possibility of detecting the adulteration of flaxseed oil samples with various concentrations of sunflower oil (10, 20, 30, 50, 70, and 90% *v*/*v*) using magnetic resonance fingerprinting (MRF) [21]. 

In Table 1, changes in DSC parameters are presented within an increasing concentration of refined rapeseed oil. An ANOVA analysis was performed to show si5gnificant differences between six concentrations of refined rapeseed oil for all the parameters measured, i.e., peak temperature (T1, T2, and T3); peak height (h1, h2, and h3); and the percentage of the peak area (P1, P2, and P3). 

Considering the changes in peak temperatures resulting from a 0 to 50% addition, it can be seen that the first peak (T1) shifted from −36 to −32 °C, the second from −31 to −25 °C, and the third peak from −25 to −19 °C. In contrast, peak height (h) did not change in the same way for all peaks. The first (h1) and third (h3) peaks increased with the addition of an adulterant, while the main peak, the second peak (h2), reduced from a value 0.6 W/g for pure flaxseed oil to 0.37 W/g for a sample with 50% of refined rapeseed oil (Table 1). Similarly, different characteristics were exhibited by the percentage of area parameter (P), where P3 was increased while P1 and P2 decreased with the addition of refined oil.

### 3.2. Changes in DSC Parameters of Flaxseed Oil Melting Phase Transition Depending on Adulterant Concentrations

Parameters determined from the DSC curve for three peaks, i.e., peak temperature (T), peak height (h), and percentage of peak area (P) versus rapeseed oil concertation were analyzed using linear regression (Figure 2). 

Linear regression analysis was also used by other authors to explain adulteration phenomena [3,30,49]. Thus, in this study, all variables were analyzed with the linear regression model to find out the trend of changes in DSC parameters resulting from the addition of the adulterant. The data in Figure 2a indicate that peak temperatures always rose to higher temperatures linearly with an increasing concentration of refined rapeseed oil added to the target oil, and it can be seen that all changes were significant (*p* ≤ 0.05). Among the three peaks, the strongest correlation was observed for the second peak, T2 (r = 0.95). In Figure 2b, the parameter of peak height (h1) for the first peak is comparatively stable (slightly increased, which is not statistically significant at that level (*p* > 0.05) upon addition of refined oils. In contrast, for the third peak, h3 increased significantly (*p* ≤ 0.05) and correlated strongly with the concentration of adulterant (r = 0.92). On the other hand, for the second peak, h2 was significantly lowered with the increment of added adulterant, with strong negative correlation, r = −0.96. The parameters for percentage of area calculated for each peak P1, P2, and P3 were also plotted against concentration of refined rapeseed oil. Clearly, the first and second peaks’ area proportion to the total melting curve area decreased significantly, while the percentage area of the third peak increased significantly (*p* ≤ 0.05), with a strong correlation to the adulterant concentration (r = 0.92).

## 4. Discussion

### 4.1. Classification Models for Predicting Cold-Pressed Flaxseed Oil Adulteration Levels

In order to assess the ability to build models that classify adulterated oil samples into appropriate classes, the LDA, MARS, SVM, and ANNs methods were used. Linear discriminant analysis (LDA) was used to build the first classification model. LDA and the related Fisher’s linear discriminant (FLD) are used in machine learning to find the linear combination of features that best distinguish between two or more classes of objects. The resulting combinations are used as a linear classifier. Discriminant analysis resulted in a statistically significant model with Wilks’ Lambda = 0.00119 and *p* ≤ 0.05. All variables (DSC parameters) except h3 have significant statistical discriminant power. From our study, five discrimination functions were obtained based on Wilks’ Lambda statistics, with *p* ≤ 0.05 for the first three functions. For the purposes of classifying the cases, six classification functions were calculated. Each classification function represents a linear equation that combines the input variables (DSC parameters) to discriminate between six groups (G_1 to G_6) and thus provides different classes (C1 to C6). The case is classified by evaluating the values of the classification functions for that case and assigning it to the class associated with the highest C value. The six classification functions are as follows:C1 = −71.3*T1−48.9*T2−127.8*T3 + 2483*h1 + 2487.6*h2 + 2172.7*h3 + 175.4*P1 + 206.1*P2 + 170*P3−13,488.8 (1)
C2 = −68.7*T1−47.6*T2−125.2*T3 + 2672.8*h1 + 2442.4*h2 + 2240.3*h3 + 175.8*P1 + 206.1*P2 + 170.4*P3−13,326.2(2)
C3 = −65.5*T1−46.6*T2−125.1*T3 + 2740.0*h1 + 2413.7*h2 + 2284.9*h3 + 177.4*P1 + 207.5*P2 + 172.2*P3−13,332.2(3)
C4 = −69.5*T1−44.8*T2−121.4*T3 + 3034.4*h1 + 2242.8*h2 + 2478.0*h3 + 176.9*P1 + 207.6*P2 + 173.2*P3−13,341.7(4)
C5 = −67.6*T1−40.9*T2−119.8*T3 + 3346.2*h1 + 2059.7*h2 + 2606.5*h3 + 175.4*P1 + 207.1*P2 + 173*P3−13,058.4(5)
C6 = −59.4*T1−52.2*T2−103.2*T3 + 3328.9*h1 + 1783.4*h2 + 2843.7*h3 + 168.8*P1 + 198.3*P2 + 168.8*P3−12,082.0(6)
where variables (i.e., T1, T2, T3, h1, h2, h3, P1, P2, and P3) are DSC parameters related to the case being classified, the constants (e.g., −71.3, −48.9, −127.8, etc.) are regression coefficients (slope), the constant term (e.g., −13,488.8, −13,326.2, etc.) represents the intercept in the linear equation.

Figure 3 presents the results of the discriminant analysis. From each classification function, the value C can be calculated based on the linear combination of the DSC variables and their corresponding coefficients. The higher the C value, the more likely the case belongs to the corresponding class. It is important to note that the coefficients in the classification functions are obtained through the LDA algorithm, which allowed the separation between classes to be maximized, based on the available data from DSC melting curves. The confusion matrix indicated (Table 2) that only one oil sample with 5% adulteration was classified as a 10% adulterated sample. Thus, the accuracy of the LDA model was 99.5%. A similar approach to detecting adulterations in peanut oil was adopted by other authors, where the identification accuracy was 97% [12].

MARS regression was used to build the second classification model. Multivariate adaptive regression splines (MAR Splines) is the implementation of a generalization of a technique introduced into wide use by Friedman [50] and used to solve both regression and classification problems. MARS is a non-parametric procedure requiring no assumptions about the functional relationship between the dependent and independent variables. MAR Splines model this relationship with a set of coefficients and so-called basis functions that are entirely determined from the data. In this study, MARS models created for the data matrix included a maximum of 21 basis functions. The penalty was set to 2, and the threshold to 0.0005. The MARS model of the first order was created for classification purposes, and the maximum number of terms was limited by pruning. The model has six basis functions and seven terms with GCV = 0.516. Increased numbers of the basis functions did not decrease the GCV error. MARS model coefficients and knots are presented in Table 3. The model developed here allows 90.3% of correct classifications to be obtained. The confusion matrix indicated that six samples were incorrectly classified. Therefore, the accuracy of the model based on MARS analysis was about 95.7%, as presented in Table 4. The MARS regression model was also used by other researchers to define the discriminant surface for studying the authentication of cod liver oil [13].

Another model that was examined for its usefulness in classifying oil samples into different adulteration classes was the support vector machines (SVM) model. SVM is a method for classifying samples on the basis of the variables (predictors) that describe them. It is a supervised technique, that is, with a supervisor, i.e., there are both variables describing the samples, and their membership is in defined classes in the learning sample. The support vector method performs classification tasks by constructing hyperplanes in a multidimensional space that separates samples belonging to different classes. For SVM model calculations, the datasets were divided into three subsets in a ratio of 2:1:1 (training, validation, and test set). Samples were classified via the C-SVM method with a linear Kernel type. As a result of learning, a model was obtained that allowed an almost 92% (Table 2) correct classification of oil samples with 97.3% accuracy (Table 4). Another study showed the classification accuracy of SVM as 96.25% while comparing chemometrics and AOCS official methods for predicting the shelf life of edible oil [51]. 

The last classification model built was an artificial neural network model (ANN). For calculating the ANN model, the datasets were divided into three subsets in a ratio of 2:1:1 (training, validation, and test set). The ANN model was trained using selected parameters from the dataset and was subsequently validated using an independent dataset. A multilayer feed-forward connected ANN was trained with the Broyden–Fletcher–Goldfarb–Shanno (BFGS) learning algorithm (200 epoch). The search for an appropriate ANN model was conducted using multilayer perceptron (MLP) and radial basis function (RBF) networks. In total, 20 networks were evaluated, and the best 5 were retained. The neural network consists of an input layer, one hidden layer, and one output layer. The network architecture, mainly the size of the hidden layer, was selected empirically, taking into consideration the accuracy of predicting the results. The best five ANN-MLP networks are presented in Table 5.

In the neural network obtained for oil sample classification, the Linear, Exponential, and Tanh functions were used in the hidden layer, while Softmax and Exponential functions were used in the output layer. In the input layer, there are nine neurons, which are DSC parameters. The number of neurons in hidden layer varies from 4 to 11, while the output layer contains six neurons representing each class of oil adulteration. A model consisting of the best five networks was used for oil sample classification. The accuracy of the resulting ANN model is almost 98% (Table 4) with only three samples misclassified (Table 2). This finding can be compared with the study conducted by Firouz et al. [52], who employed the classification and quantification of sesame oil adulteration and acquired 100% accuracy. 

On the basis of the models’ performance parameters, it was determined that the best model is the LDA model, with the highest values for accuracy (99.5%), precision (98.4%), sensitivity (98.4%), specificity (99.7%), and F1-score (98.4%), and the lowest value of misclassification rate equals 0.54%. In contrast, the worst one was the MARS model, which had the lowest values for accuracy (95.7%), precision (87%), sensitivity (87%), specificity (97.4%), and F1-score (87%), and the highest value of misclassification rate equals 4.3%. The second-best model was the ANN model, and the third was the SVM model. The accuracy of all these models was very high, which suggested its ability to predict adulterated oil samples into appropriate classes.

### 4.2. Regression Models for Predicting the Concentration of Refined Rapeseed Oil in Cold-Pressed Flaxseed Oil

Multiple regression analysis (MLR) was performed to formulate a general linear equation that would fit the variables from DSC melting curves against different concentrations of adulterants. This would provide the possibilities to detect the percentage of adulterants in any sample. The MLR model that was obtained was statistically significant with F (9.52) = 364.57 (*p* ≤ 0.05), R^2^ = 0.9844, and adjusted R^2^ = 0.9817. The standard error of estimation was 2.3028.

Table 6 demonstrates the summary of DSC parameters, where (b*) values refer to the standardized regression coefficient, and (b) values refer to the regular regression coefficient. Determining (b*) allowed for a direct comparison of the magnitude and importance of the independent variables, where we can see the highest values are presented for h2, T3, and h3 as −0.32, 0.27, and 0.16, respectively. On the other hand, (b) values signify the slope coefficient associated with an independent variable. It represents the change in the dependent variable for a one-unit change in the corresponding independent variable while holding all other independent variables constant. Table 6 shows that the h2 (−65.52) variable has the strongest negative relationship with the concentration variables, indicating that with a decreased value of h2, the concentration of adulterants increased. On the other hand, h3 and T2 variables consequently increased or decreased linearly with the concentration values of adulterants. Accordingly, a model with statistically significant predictors was built.
% adulterant = 2.185*T3 − 109.995*h2 + 96.576*h3 + 105.319 ± 2.567(7)
where T3 represents the third strongest significant independent variable (*p* = 0.000), h2 represents the highest strongest significant independent variable (*p* = 0.000), and h3 represents the second strongest significant independent variable (*p* = 0.000).

The goodness of fit of the model to the experimental data and the coefficient of determination R2 and the coded coefficient of determination were 0.978 and 0.977, respectively. Equation no. 7 can be used to estimate the percentage of adulterants (for this study, refined rapeseed oil) in a cold-pressed flaxseed oil sample based on three dependent variables: T3, h2, and h3. The equation implies that the variables T3, h2, and h3 are assumed to have a linear relationship with a percentage of the adulterant. The correlation between observed and predicted values was 0.992 with a low RMSE value of 2.12 (Table 7). A similar study by Sim et al. [2] showed that it was possible to predict adulteration of lard in palm oil olein using the MLR model, where the prediction performance was measured based on the percentage root mean square error (%RMSE).

MARS regression was used to build the second regression model. In this study, MARS models created for the data matrix included a maximum of 21 basis functions. The penalty was set to 2 and the threshold to 0.0005. The MARS model of the first order was created for classification purposes, and the maximum number of terms was limited by pruning. The model has 10 basis functions and 11 terms with GCV = 6.252. Equation (8) represents the MARS model for predicting the concentration of refined oil in the samples.
% adulterant = 1.955e − 1.9203*max(0; T2 + 2.763e) − 3.192*max(0; −2.763e − T2) + 3.6434*max(0; T1 + 3.437e) + 4.572*max(0; T3 + 2.254e) + 3.465e^2^*max(0; h1 − 1.241e^−1^) − 4.490e^2^*max(0; h1 − 1.4729e^−1^) + 2.214*max(0; 3.240e − P2) − 7.047e^−1^*max(0; P1 − 2.769e) + 4.543e^−1^*max(0; 2.769e − P1) − 7.618e*max(0; h2 − 4.510e^−1^)(8)

The correlation between the observed and predicted value was 0.995 with a low RMSE value of 1.65 (Table 7).

Another model that was examined for its usefulness in predicting oil sample adulteration was the support vector machines (SVM) model. SVM can be used for both classification and regression problems. In SVM regression, the search is for a functional dependence of the dependent variable y (% of adulteration) on a set of independent variables x (DSC parameters). For calculating the SVM model, the datasets were divided into three subsets in a ratio of 2:1:1 (training, validation, and test set) for model regression type 1 (C = 10.000000, epsilon = 0.100000) with radial basis function (gamma = 0.111111) kernel type. Samples were classified using the C-SVM method with linear kernel type. The correlation between the observed and predicted values was 0.992 with a low RMSE value of 2.1 (Table 7).

The last regression model built was an artificial neural network model (ANN). For calculating the ANN model, the datasets were divided into three subsets in a ratio of 2:1:1 (training, validation, and test set). The ANN was trained using selected parameters from the dataset and was subsequently validated using an independent dataset. A multilayer feed-forward connected ANN was trained with the Broyden–Fletcher–Goldfarb–Shanno (BFGS) learning algorithm (200 epoch). The search for an appropriate ANN model was performed using multilayer perceptron (MLP) and radial basis function (RBF) networks. In total, 20 networks were evaluated, and the best 5 were retained. The neural network consists of an input layer, one hidden layer, and one output layer. The network architecture, mainly the size of the hidden layer, was selected empirically, taking into consideration the accuracy of the results prediction. The five best ANN-MLP networks are presented in Table 8.

In the neural network obtained for predicting oil adulteration, the Logistic and Tanh functions were used in the hidden layer, while Logistic, Tanh, and Exponential functions were used in the output layer. In the input layer, there are nine neurons, which are DSC parameters. The number of neurons in the hidden layer varies from 9 to 13, while the output layer contains 14 neurons representing the refined oil concentration. A model consisting of the best five networks was used for prediction purposes. The correlation between the observed and predicted values was 0.996 with a low RMSE value of 1.51 (Table 7). The study found that the ANN regression analysis demonstrated robust models for adulteration phenomena in sesame oil generated using sunflower oil, canola oil, and sunflower + canola oils, quantitatively [52]. Another study stated that using ANN as a pattern recognition technique for the data obtained from electronic nose could not detect the proportion of adulteration in camellia seed oil but successfully quantified adulteration in sesame oil [53].

Table 7 presents the goodness of fit parameters of the regression models obtained. The best model is based on ANN algorithms and exhibits the highest R (0.996), R2 (0.992), adjusted R2 (0.922) values with the lowest values of AIC (233), BIC (240), and RMSE (1.51). The MARS model was next, followed by the SVM, and the least fitting was the MLR model.

#### Principle Component Analysis (PCA) and Orthogonal Partial Least Squares Discriminant Analysis (OPLS-DA)

This adulteration detection study involves analyzing multiple variables of DSC parameters simultaneously. Chemometric techniques like PCA and OPLS-DA are designed to handle multivariate data, allowing for a comprehensive analysis of the oil samples. For instance, PCA presented in Figure 4a reduced the dimensionality of the dataset by transforming the variables into a smaller set of principal components, capturing the most important variations in the data. Hence, as an unsupervised method, PCA represents the combinations of the original variables and can be difficult to interpret in the context of class separation. To solve this issue, OPLS-DA analysis was adopted for assessing the discrimination and classification of the adulterated flaxseed oil samples. As a fast and efficient screening tool for large datasets, OPLS-DA allowed us to evaluate the effectiveness of DSC melting profiles of adulterated flaxseed oils in classifying and detecting the percentage of adulterants concentration by differentiating them (Figure 4b). 

In Figure 4a, the DSC data matrix serves as the basis for conducting PCA analyses, which provided a visual representation of the data pattern for six concentrations of adulterants mixed with pure flaxseed oils. In the score plot, each point represents a sample (a specific concentration of adulterant mixed with flaxseed oil) in the space of two principal components, t[1] and t[2], which were able to explain 91.1% of the variation of the normalized heat flow results. Additionally, R^2^X (cum) and Q^2^ (cum) values (Table 9) are the quantities useful for PC model diagnostics as the fraction of the explained variation R^2^X and the fraction of predicted variation Q^2^. The more significant a principal component, the closer its R^2^X and R^2^X (cum) will be to value 1 for a PC model with a sufficient number of components. For the PCA analysis presented in Figure 4a, the R^2^X (cum) value was 0.973, which indicates that the retained principal components capture a larger proportion of the overall variation in the dataset. This finding can help in determining the appropriate number of components to retain for further analysis. Besides this, the Q^2^ (cum) value was 0.897 for the PCA model, which shows that the cumulative sum of the cross-validated predictive ability is high for the variables of the normalized heat flow of phase transition curves. This approach of employing PCA analysis can be compared to other studies, where researchers detected (with 100% accuracy) adulteration of flaxseed oil with rapeseed, corn, peanut, sunflower seed, soybean, and sesame oils [23], or adulteration of virgin coconut oil with refined coconut oil [14].

The next chemometric approach was analyzing the dataset of multiple variables using OPLS-DA, which can effectively enhance the separation of classes while maintaining the predictive power of the model by utilizing orthogonal projection in the score plot. The analysis aims to classify and distinguish different concentrations of adulterants (ranging from 0% to 50%) added to pure flaxseed oil. The model consists of 15 variables, where a total of nine DSC parameters are considered as X variables and six different concentrations of adulterants added are considered as Y variables representing the six classes. Five predictive components (P1 to P5) capture the between-class variation, meaning they account for the differences between the different concentrations of adulterants. The orthogonal components capture the within-class variation, representing similarities within each concentration group. Within the framework of the OPLS-DA model, the systematic variation in data was described by two distinct components. The first component, known as the predictive component, exhibits a linear relationship with the classes (Y) and possesses the ability to make accurate predictions. In Figure 4b, the *x*-axis represents the first component (t1 = 72.3%), and the Y-axis the second principal component [t2 = 11.4%]. The observations in the scatter plot are colored, based on their class, which corresponds to the different concentrations of adulterants added to pure flaxseed oil. The scatter plot serves as a visualization of how the modeled observations in the X space are positioned relative to each other. Observations that are close to each other in the plot indicate a higher degree of similarity compared to those that are farther apart.

Also, in Table 9, the R^2^X (cum) value was presented as 0.986, indicating that the OPLS-DA model fits the X data well, capturing a large portion of the variation present in the DSC parameters. On the other hand, a Q^2^ (cum) value of 0.33 indicates that the OPLS-DA model can predict approximately 33% of the variation in the Y data, according to cross-validation. The range of Q^2^ values suggests that the model has reasonable predictive ability for the concentration of adulterants based on DSC variables. OPLS-DA was also adopted by other authors to determine important variables when detecting flaxseed oil multiple adulteration via near-infrared spectroscopy. These authors also adopted the one-class partial least squares (OCPLS) method to build a detection model, which provided a high accuracy of 95.8% [15].

Although the model demonstrates a good fit to the X data (DSC parameters), a low Q^2^ (cum) value (0.33) indicates that the model’s ability to explain and predict the variation for the Y data (adulterant concentrations) was poor. Thus, the authors decided to explore an alternative modeling technique, i.e., the partial least squares (PLS) technique. In Figure 5a, a loading plot of PLS analysis is presented for DSC parameters obtained from the melting curves. To obtain a comprehensive understanding of the model’s performance and predictive ability, the R^2^X (cum) value was calculated at a level of 0.953 and was lower than for OPLS-DA, while the predictive Q^2^ (cum) value was higher than for OPLS-DA (0.973). The results obtained for R2X (cum) and Q^2^ (cum) indicate that the PLS model had higher predictive power than OPLS-DA. Additionally, Figure 5b presents the variables’ influence on the projection (VIP) plot, which provides information about the importance of variables (DSC parameters) that are above value 1. As was the case with the MLR model (Table 6), the parameter for the first peak height (h1) and percentage area (P1) were not significant.

In addition to the approaches presented in the study, the observed and predicted value graph from the PLS model is presented in Figure 6. By plotting the observed concentrations of adulterants (actual values) against the predicted concentrations (values predicted using the PLS model) on a graph, it was possible to assess visually how well the model predicts the adulterant levels in the flaxseed oil samples, based on the DSC parameters from melting curves. We can see that the observed and predicted values align closely along a diagonal line, which indicates that the PLS model accurately predicts the adulterant concentrations based on the DSC parameters. A Pearson’s correlation coefficient (r) of 0.995 between the observed and predicted values indicates an extremely strong positive linear relationship between the two sets of values. This graph also shows that this model can effectively differentiate between pure flaxseed oil and adulterated samples, providing a reliable means of detecting and estimating the adulterant concentrations. By assessing this graph, it is also evident that the PLS model has successfully learned the relationship between the DSC parameters and the adulterant concentrations, which validates the model for this purpose. This finding can be compared with the study conducted by Rocha et al., who adopted the PLS method for the classification and quantification of different types of blended biodiesel synthesized from peanut, corn, and canola oils and observed a Pearson’s correlation coefficient of 0.969 between the real and predicted concentrations [11].

## 5. Conclusions

By developing a method for detecting adulteration of flaxseed oil, this study contributes to solving the problem of food adulteration and its economic impact on the global food industry. The DSC melting curves provided unique and substantial information about the thermal behavior of the flaxseed oil and showed distinct changes when adulterants were added. The second peak in the DSC profiles was identified as the major peak, and its characteristics, such as peak temperature, peak height, and percentage of peak area, were found to be significantly affected by the concentration of adulterants. Moreover, the findings demonstrate the efficacy of coupling DSC with chemometric methods in detecting and classifying adulterations in cold-pressed flaxseed oil. Of the classification models built, the LDA model exhibited the best performance, underlining its potential for accurate identification of adulterated oil samples. On the other hand, the regression model based on the ANN algorithm showed the best goodness of fit for DSC parameters regarding the prediction of adulterant concentrations. The equation and PMML codes derived from the MLR, MARS, SVM, and ANN regression analyses can be used to estimate the percentage of adulterants in flaxseed oil samples based on the values of peak temperature, peak height, and percentage of peak area. The study also employed other chemometric techniques, such as PCA, OPLS-DA, and PLS to effectively classify and describe the adulteration phenomena. The resulting plots demonstrated that the PLS model showed the greatest accuracy in predicting adulterant levels in flaxseed oil samples based on DSC parameters, as indicated by a strong positive linear relationship (Pearson’s correlation coefficient of 0.995). The PLS model effectively differentiated between pure flaxseed oil and adulterated samples, providing a reliable means of detecting and estimating adulterant concentrations. This study highlights the significance of combining DSC with chemometric methods for detecting adulterations in flaxseed oil and emphasizes the importance of quality assessment and authenticity verification in the food industry.

## Figures and Tables

**Figure 1 foods-12-03352-f001:**
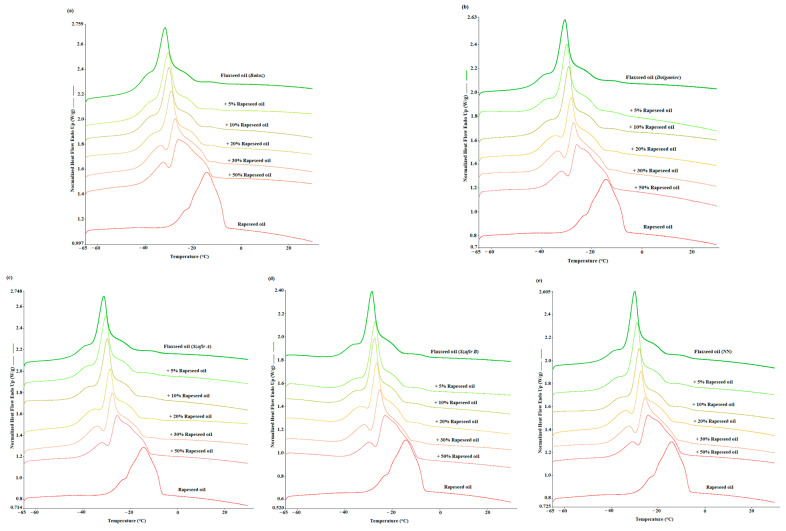
DSC melting curves obtained at a 5 °C/min heating rate for different cultivars of cold-pressed flaxseed oils adulterated with refined rapeseed oils in different concentrations 0, 5, 10, 20, 30, and 50% *w*/*w*. (**a**) *Bukoz* cultivar, (**b**) *Dolguniec* cultivar, (**c**) *Szafir A* cultivar, (**d**) *Szafir B* cultivar, and (**e**) unknown cultivar.

**Figure 2 foods-12-03352-f002:**
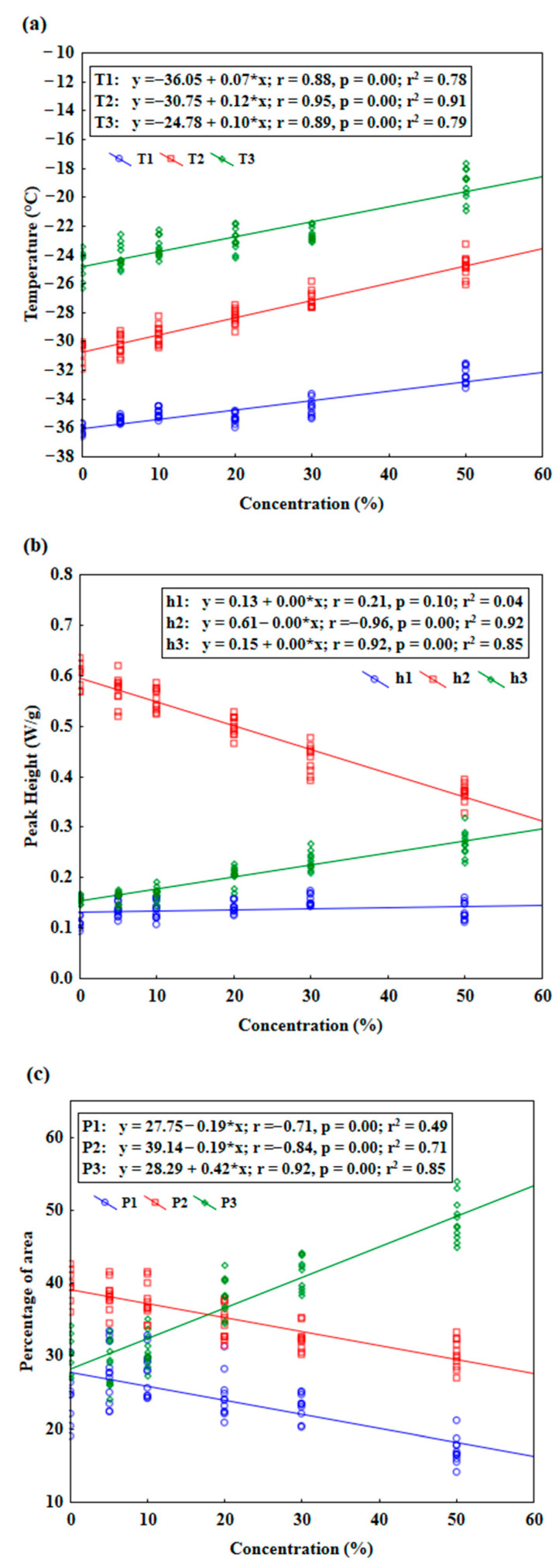
Regression analysis of adulterated flaxseed oil samples for DSC parameters. (**a**) Peak temperature (T, °C), (**b**) peak height (h, W/g), and (**c**) percentage of area (P).

**Figure 3 foods-12-03352-f003:**
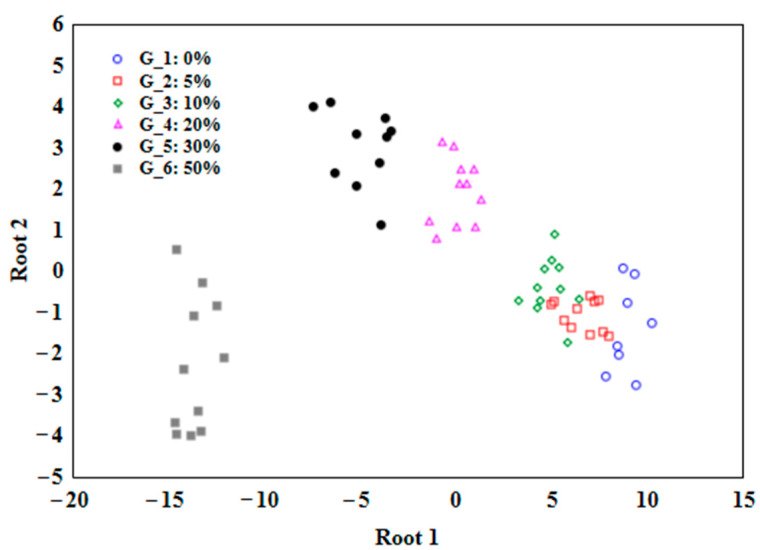
Linear discrimination analysis plot (LDA) for cold-pressed flaxseed oil adulterated with various concentrations of refined rapeseed oils (0, 5, 10, 20, 30, and 50% *w*/*w*).

**Figure 4 foods-12-03352-f004:**
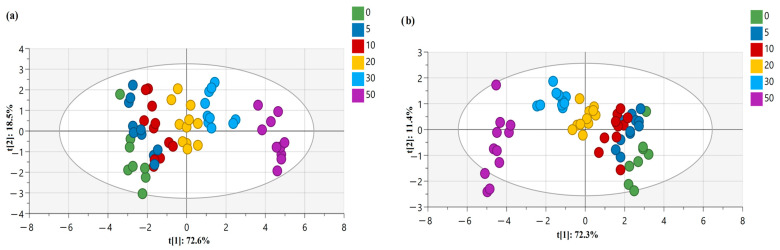
Score plots obtained via (**a**) PCA and (**b**) OPLS-DA for cold-pressed flaxseed oil adulterated with various concentrations of refined rapeseed oils (0, 5, 10, 20, 30, and 50% *w*/*w*).

**Figure 5 foods-12-03352-f005:**
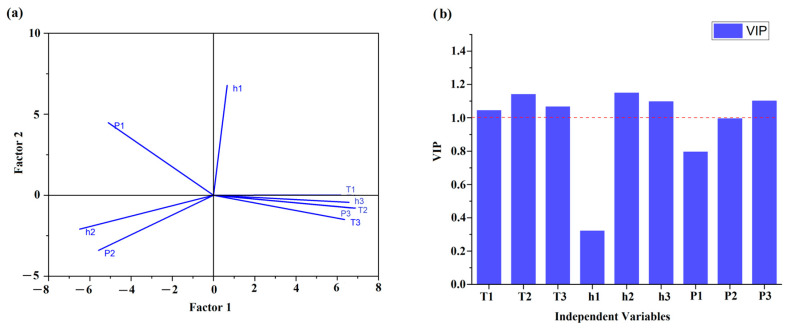
Loading plot of (**a**) PLS analysis for all DSC parameters determined for cold-pressed flaxseed oils adulterated with various concentrations of refined rapeseed oil (0, 5, 10, 20, 30, and 50% *w*/*w*). (**b**) The variables’ influence on the projection (VIP) graph.

**Figure 6 foods-12-03352-f006:**
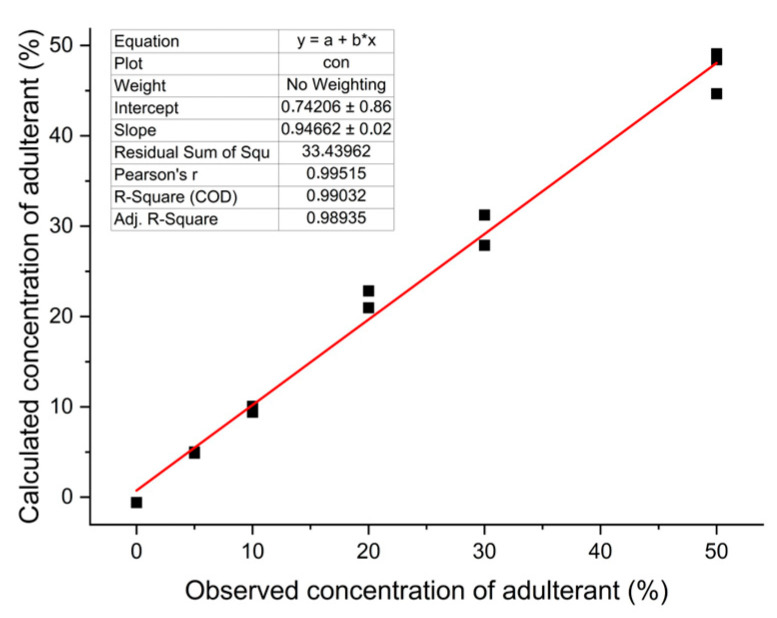
Observed and predicted values in a partial least squares (PLS) model based on the DSC parameters.

**Table 1 foods-12-03352-t001:** DSC thermodynamic parameters of melting phases for cold-pressed flaxseed oils adulterated with different concentrations of refined rapeseed oils.

DSC Parameters	Concentrations					
	0%	5%	10%	20%	30%	50%
T1	−36.15 ± 0.35 ^a^	−35.44 ± 0.23 ^bc^	−34.95 ± 0.35 ^bc^	−35.32 ± 0.36 ^b^	−34.57 ± 0.59 ^c^	−32.39 ± 0.6 ^d^
T2	−30.67 ± 0.69 ^a^	−30.26 ± 0.67 ^ab^	−29.55 ± 0.68 ^b^	−28.31 ± 0.54 ^c^	−27.08 ± 0.61 ^d^	−24.78 ± 0.77 ^e^
T3	−24.73 ± 1.04 ^a^	−24.08 ± 0.84 ^ab^	−23.44 ± 0.72 ^bc^	−22.92 ± 0.81 ^c^	−22.53 ± 0.48 ^c^	−19.12 ± 1.07 ^d^
h1	0.12 ± 0.02 ^a^	0.14 ± 0.02 ^abc^	0.14 ± 0.02 ^abc^	0.14 ± 0.01 ^bc^	0.15 ± 0.01 ^c^	0.13 ± 0.02 ^ab^
h2	0.60 ± 0.03 ^e^	0.57 ± 0.03 ^de^	0.55 ± 0.02 ^d^	0.50 ± 0.02 ^c^	0.44 ± 0.03 ^b^	0.37 ± 0.02 ^a^
h3	0.16 ± 0.01 ^a^	0.16 ± 0.01 ^a^	0.17 ± 0.01 ^a^	0.21 ± 0.02 ^b^	0.23 ± 0.02 ^c^	0.27 ± 0.02 ^d^
P1	24.12 ± 3.58 ^bc^	27.50 ± 4.00 ^c^	27.55 ± 3.17 ^c^	24.41 ± 3.04 ^bc^	23.09 ± 1.74 ^b^	17.08 ± 1.84 ^a^
P2	39.77 ± 2.16 ^c^	38.32 ± 1.96 ^c^	37.49 ± 2.51 ^c^	34.52 ± 2.35 ^b^	32.43 ± 1.69 ^ab^	30.27 ± 2.00 ^a^
P3	30.49 ± 2.57 ^a^	28.78 ± 3.07 ^a^	30.75 ± 2.31 ^a^	38.30 ± 2.52 ^b^	41.50 ± 2.26 ^b^	48.71 ± 2.96 ^c^

All values are mean ± standard deviation (*n* = 10), (a–e)—means with the same letters within the column are not different (*p* > 0.05). T1, T2, and T3 represent the first, second, and third peak temperatures, respectively; h1, h2, and h3 means peak height for the first, second, and third peak, respectively; P1, P2, and P3 represent percentage of peak area for first, second, and third peak, respectively.

**Table 2 foods-12-03352-t002:** Confusion matrix of cold-pressed flaxseed oils adulterated with different concentrations of refined rapeseed oils.

	Observed	0%	5%	10%	20%	30%	50%
Predicted							
0%	LDA	8	0	0	0	0	0
	ANN	8	0	0	0	0	0
	SVM	6	2	0	0	0	0
	MARS	7	1	0	0	0	0
5%	LDA	0	10	1	0	0	0
	ANN	0	10	1	0	0	0
	SVM	0	10	1	0	0	0
	MARS	0	9	2	0	0	0
10%	LDA	0	0	11	0	0	0
	ANN	0	3	8	0	0	0
	SVM	0	2	9	0	0	0
	MARS	0	1	10	0	0	0
20%	LDA	0	0	0	11	0	0
	ANN	0	0	0	11	0	0
	SVM	0	0	0	11	0	0
	MARS	0	0	0	9	2	0
30%	LDA	0	0	0	0	10	0
	ANN	0	0	0	0	10	0
	SVM	0	0	0	0	10	0
	MARS	0	0	0	2	8	0
50%	LDA	0	0	0	0	0	11
	ANN	0	0	0	0	0	11
	SVM	0	0	0	0	0	11
	MARS	0	0	0	0	0	11

**Table 3 foods-12-03352-t003:** MARS model coefficients and knots calculated for classification of cold-pressed flaxseed oils adulterated with different concentrations of refined rapeseed oils.

	Intercept	Term 1	Term 2	Term 3	Term 4	Term 5	Term 6
0%	−1.01	−3.39	−9.46 × 10^−1^	8.85 × 10^−1^	2.39 × 10^−2^	9.37 × 10^−1^	8.68
5%	1.27	−1.39	1.35	−9.96 × 10^−1^	3.12 × 10^−1^	−1.28	4.00
10%	−1.02 × 10^−1^	1.81 × 10^1^	−3.68 × 10^−1^	−2.01 × 10^−1^	−7.60 × 10^−2^	2.72 × 10^−1^	−2.12 × 10^1^
20%	3.38 × 10^−1^	2.92	1.83 × 10^−1^	3.18 × 10^−1^	−3.07 × 10^−1^	−2.26 × 10^−1^	−9.92
30%	5.08 × 10^−1^	−1.58 × 10^1^	−6.08 × 10^−1^	−1.67 × 10^−2^	4.51 × 10^−2^	2.55 × 10^−1^	1.79 × 10^1^
50%	1.39	−2.29	5.25 × 10^−1^	−1.03 × 10^−1^	3.13 × 10^−2^	−8.92 × 10^−2^	4.43
Knots T1			−3.44 × 10^1^	−3.44 × 10^1^		−3.55 × 10^1^	
Knots T2					−2.88 × 10^1^		
Knots h2		4.83 × 10^−1^					5.28 × 10^−1^

**Table 4 foods-12-03352-t004:** Performance parameters of models for classification of cold-pressed flaxseed oils adulterated with different concentrations of refined rapeseed oils.

Performance Parameter	Accuracy	Misclassification Rate	Precision	Sensitivity	Specificity	F1-Score
Model						
LDA	99.46%	0.54%	98.39%	98.39%	99.68%	98.39%
ANN	97.85%	2.15%	93.55%	93.55%	98.71%	93.55%
SVM	97.31%	2.69%	91.94%	91.94%	98.39%	91.94%
MARS	95.70%	4.30%	87.10%	87.10%	97.42%	87.10%

**Table 5 foods-12-03352-t005:** ANN models calculated for the classification of cold-pressed flaxseed oils adulterated with different concentrations of refined rapeseed oils.

Net Architecture	Training Accuracy	Test Accuracy	Validation Accuracy	Training Algorithm	Error Function	Hidden Activation	Output Activation
MLP 9-9-6	88.636	100.000	77.778	BFGS 10	Entropy	Linear	Softmax
MLP 9-11-6	88.636	88.889	100.000	BFGS 11	Entropy	Linear	Softmax
MLP 9-8-6	81.818	88.889	100.000	BFGS 9	Entropy	Linear	Softmax
MLP 9-9-6	84.091	77.778	100.000	BFGS 32	SOS	Exponential	Exponential
MLP 9-4-6	84.091	88.889	88.889	BFGS 15	Entropy	Tanh	Softmax

**Table 6 foods-12-03352-t006:** Summary of independent variables in multiple regression analysis.

DSC Parameters	b* (Standardized Co-Efficient)	b (Raw Co-Efficient)	*p*-Value
		145.3464 *	0.000059 *
T1	0.090680	1.2225	0.126748
T2	−0.052531	−0.4175	0.627294
T3	0.273274 *	2.3374 *	0.000332 *
h1	0.080773	73.6583	0.056670
h2	−0.324189 *	−65.5228 *	0.000014 *
h3	0.168666 *	65.1633 *	0.004121 *
P1	−0.128623	−0.4718	0.151392
P2	−0.145236	−0.6314	0.083058
P3	0.019714	0.0435	0.896428

* Coefficients are significant statistically (*p* ≤ 0.05).

**Table 7 foods-12-03352-t007:** Goodness of fit parameters between observed and predicted regression models for the prediction of concentrations of refined rapeseed oils in cold-pressed flaxseed oil.

Model	R	R²	Adjusted R^2^	AIC	BIC	RMSE
ANN	0.996	0.992	0.992	233	240	1.51
MARS	0.995	0.990	0.990	244	251	1.65
SVM	0.992	0.985	0.984	274	280	2.10
MLR	0.992	0.984	0.984	275	281	2.12

**Table 8 foods-12-03352-t008:** ANN models calculated for the prediction of refined rapeseed oil concentrations in cold-pressed flaxseed oils.

Net Architecture	Training Accuracy	Test Accuracy	Validation Accuracy	Training Algorithm	Error Function	Hidden Activation	Output Activation
MLP 9-9-1	0.9957	0.9919	0.9901	BFGS 43	SOS	Logistic	Logistic
MLP 9-13-1	0.9964	0.9961	0.9941	BFGS 46	SOS	Logistic	Tanh
MLP 9-11-1	0.9965	0.9913	0.9885	BFGS 61	SOS	Logistic	Exponential
MLP 9-13-1	0.9961	0.9899	0.9866	BFGS 54	SOS	Logistic	Exponential
MLP 9-11-1	0.9963	0.9937	0.9972	BFGS 37	SOS	Tanh	Tanh

**Table 9 foods-12-03352-t009:** Summary of the fit of different models for discrimination (differentiation) of samples with different concentrations of adulterants (refined rapeseed oil).

Model	R^2^X (cum)	R^2^ (cum)	Q^2^ (cum)
PCA	0.973		0.897
OPLS-DA	0.986	0.465	0.330

## Data Availability

The data presented in this study are available upon reasonable request.

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
