# Peer review of "Comparing Different Chemometric Approaches to Detect Adulteration of Cold-Pressed Flaxseed Oil with Refined Rapeseed Oil Using Differential Scanning Calorimetry"

_foods, 2023, doi:10.3390/foods12183352_

Round 1

Reviewer 1 Report (Previous Reviewer 1)

I thank the authors for their efforts to correct the manuscript. Compared with the previous food adulteration, it is easier to understand with chemometric approaches as the main content of the research. However, the selection of quality indicators for food adulteration still needs to be clearly described and explained.

Author Response

Reviewer 1:

I thank the authors for their efforts to correct the manuscript. Compared with the previous food adulteration, it is easier to understand with chemometric approaches as the main content of the research. However, the selection of quality indicators for food adulteration still needs to be clearly described and explained.

Response:

We would like to thank the Reviewer for the effort put into the whole review process and finaly giving a positive opinion. Regarding the selection of quality indicators, in this study on the adulteration of flaxseed oil, it was based on a thorough analysis of the DSC melting curves. First step in the determination of DSC quality parameters was the analysis of flaxseed oil melting curves. Due to complexity of this curve, caused by the overlapping peaks, it was necessary to implement the procedure of peaks deconvolution to determine more accurately the parameters of each peak. Peaks deconvolution enabled to separate individual peaks and to determine their corresponding parameters more accurately. Upon analyzing the deconvoluted curves, it became evident that the parameters associated with the peaks were responsive to the addition of the adulterant (refined rapeseed oil). The important parameters describing peaks included: peak temperature, peak height and peak area. All of them provided insights into the alterations induced by the presence of adulterants in the flaxseed oil. Upon analyzing the deconvoluted curves, it became evident that the parameters associated with the peaks were responsive to the addition of the adulterant (refined rapeseed oil). The parameter of peak temperature (T1, T2, T3) defines peak position in DSC curve and it has changed with the addition of the adulterant. These temperature shifts directly reflect changes in the composition of the sample. For instance, the position of the first peak (T1) shifted from –36 to –32 °C, T2 shifted from –31 to –25 °C, and T3 shifted from –25 to –19 °C as the adulterant concentration increased. The complementary evaluation was carried out by peak height and area, which have expressed changes in the shape of curve due to the addition of adulterants. These two parameters complement each other by capturing different aspects of the alterations in the melting curves. It is noteworthy that similar changes in the thermal transition profile were observed across various cultivars of flaxseed oil when adulterated with refined rapeseed oil. This consistency in the response further reinforces the robustness of the selected quality indicators. Based on these findings, selected DSC indicators provide comprehensive and reliable data on the fingerprint of the genuine flaxseed oil as well as adulterated  oil as they capture variations in peak positions, shapes, and relative proportions that are directly influenced by the adulterant concentration.

Reviewer 2 Report (Previous Reviewer 2)

Paper has been much improved since last time. Although I did not manage to see earlier comments I believe it is quite innovative since the authors developed a method for detecting adulteration of flaxseed oil, thus

contributing to solving the problem of food adulteration and its economic impact on the global food industry by employing DSC melting curves.

Author Response

Reviewer 2:

Paper has been much improved since last time. Although I did not manage to see earlier comments I believe it is quite innovative since the authors developed a method for detecting adulteration of flaxseed oil, thus contributing to solving the problem of food adulteration and its economic impact on the global food industry by employing DSC melting curves.

Dear Reviewer,

We would like to thank for taking the time and effort put into the review of our manuscript. We appreciate all insightful comments and suggestions given throughout the whole review process.

Reviewer 3 Report (Previous Reviewer 4)

This article is interesting, some comments to improve this article are:

1. Abstract is good

2. In introduction, the gap analysis and the importance of different use of chemometrics techniques must be highlight.

3. The novelty of this study has been stated.

4. The relevant reference should be cited in Methods.

5. the basic principle of each chemometrics should be explained briefly in Methods

6. The validity model should be assessed in terms of overfitting should be explained

Author Response

Dear Reviewer,

We would like to thank for taking the time and effort put into the review of our manuscript. We appreciate all insightful comments and suggestions, which we have carefully considered and corrected. Our responses are listed below:

This article is interesting, some comments to improve this article are:

  1. Abstract is good
  2. In introduction, the gap analysis and the importance of different use of chemometrics techniques must be highlight.

Answer: The following sentence was added to highlight-

Line 77-86: “In pursuit of the idea of gap analysis, DSC stands out as an analytical method with the ability to detect changes associated with changes in the composition of triacylglycerols, which makes it possible to use it as a "at-a-glance" method for oil authentication. This method measures the thermodynamic parameters of temperature and enthalpy of phase transition without the use of any chemicals, which is not possible in the case of liquid chromatography. Unlike other methods like FTIR [24], XRD [35], and NMR [14], the thermal behavior of the material can be studied under different conditions, e.g. scanning rate. It has already been proven ability of DSC technique to provide quantitative thermal data in fields such as pharmaceuticals [36], polymers [37], and food science [38].”

Line 55-63: “Researchers have shown the successful application of combining analytical techniques with linear and non-linear chemometric tools [9] to build classification and regression models for oil samples, which clearly demonstrates the importance of different uses of chemometric techniques, i.e., linear discriminant analysis (LDA)[10], multiple linear regression (MLR)[2], multivariate adaptive regression splines (MARS) [11], support vector machine (SVM) [12], artificial neural networks (ANNs) [13], principle component analysis (PCA) [14], orthogonal partial least squares discriminant analysis (OPLS-DA) [15] and partial least squares regression (PLS) [16].”

  1. The novelty of this study has been stated.
  2. The relevant reference should be cited in Methods.

Answer: Following reference was added to the methods:

  1. American Oil Chemists’ Society. DSC melting properties of fats and oils. Cj 1-94. Illinois, USA; 2000.
  2. 5. the basic principle of each chemometrics should be explained briefly in Methods

Answer:

All chemometric methods used are briefly described in the results section:

Line: 286-290: Linear discriminant analysis (LDA) was used to build the first classification model. LDA and the related Fisher's linear discriminant (FLD) are used in machine learning to find the linear combination of features that best distinguish between two or more classes of objects. The resulting combinations are used as a linear classifier.

Line: 321- 327: Multivariate Adaptive Regression Splines (MAR Splines) is the implementation of a generalization of a technique introduced into wide use by Friedman [49] and used to solve both regression and classification problems. MARS is a non-parametric procedure requiring no assumptions about the functional relationship between the dependent and independent variables. MAR Splines models this relationship with a set of coefficients and so-called basis functions that are entirely determined from the data.

Line: 343-348: SVM is a method for classifying samples on the basis of the variables (predictors) that describe them. It is a supervised technique, that is, with a supervisor, i.e., there are both variables describing the samples and their membership in defined classes in the learning sample. The support vector method performs classification tasks by constructing hyperplanes in a multidimensional space that separates samples belonging to different classes.

Line: 357-361: The ANN model was trained using selected parameters from the data set and was subsequently validated using an independent data set. Multilayer feed-forward connected ANN was trained with the Broyden-Fletcher-Goldfarb-Shanno (BFGS) learning algorithm (200 epoch). The search for an appropriate ANN model was done using multilayer perceptron (MLP) and radial basis function (RBF) networks.

Line: 389-392: Multiple regression analysis (MLR) was performed to formulate a general linear equation which will fit the variables from DSC melting curves against different concentrations of adulterants. This will provide the possibilities to detect the percentage of adulterants in any sample.

However, in the methods section the following sentence was added with relevant references:

Line: 164-166: Classification and regression approaches were used to build predictive models of oil adulteration as it was proposed by authors for the food analysis purpose [9][48].

Also, in the introduction section, all methods were mentioned with relevant literature references:

Line 58-63: linear discriminant analysis (LDA)[10], multiple linear regression (MLR)[2], multivariate adaptive regression splines (MARS) [11], support vector machine (SVM) [12], artificial neural networks (ANNs) [13], principle component analysis (PCA) [14], orthogonal partial least squares discriminant analysis (OPLS-DA) [15] and partial least squares regression (PLS) [16].

  1. The validity model should be assessed in terms of overfitting should be explained

Answer: Due to the small amount of data, the authors decided to use Leave-One-Out Cross-Validation (LOOCV). Leave-One-Out Cross-Validation (LOOCV) is a specific type of cross-validation where the number of folds equals the number of data points in the dataset. In other words, for a dataset with N samples, LOOCV will have N folds. For each fold, just one data point is used for testing while the remaining N-1 data points are used for training. The process is repeated N times, each time using a different data point as the test set. Since LOOCV uses all but one sample for training, the model gets trained on a dataset that is very similar to the original. This results in low bias, helping the model to capture the underlying trend in the data accurately. LOOCV provides a robust assessment of model performance because it averages the performance over all individual samples in the dataset. This comprehensive evaluation can be useful for understanding how well the model generalizes. However, the authors are aware of the limitations of this validation method. LOOCV often exhibits high variance in the performance estimates because each training set is almost identical to the others. This similarity can lead the model to overfit to the training set. Therefore, the authors performed another 10-fold cross-validation, which showed a very similar model fit. Probably because the DSC results are consistent and not noisy. Therefore, the authors finally used LOOCV validation, which is more efficient with a small dataset. Of course, the paper is a prelude to building a model that will be built using more data, in which case the authors plan to use external validation.

This manuscript is a resubmission of an earlier submission. The following is a list of the peer review reports and author responses from that submission.

Round 1

Reviewer 1 Report

The authors are very creative in using the differential scanning calorimetry (DSC) as a source for constructing the chemometric database. However, the integrity of the system still needs to be considered, and the current research design framework is too brief to mislead readers. Can variability in flaxseed oil samples be conserved? Can any flaxseed oil be distinguished from other oil types blended (adulterated)? The conservative nature of refined rapeseed oils should also be evaluated first, and there should even be several random ratios of rapeseed oils mixed with flaxseed oils.

Under the current research design, it is still correct to replace the title "Using chemometric approaches with DSC to detect spiking of cold-pressed flaxseed oils in a refined rapeseed oil" with the opposite meaning.

Reviewer 2 Report

This is an original article entitled Different chemometric approaches to detect adulteration of cold-pressed flaxseed oil with refined rapeseed oil using differential scanning calorimetry. 

three chemometric approaches were compared: 1) classification models (Linear Discriminant Analysis, LDA Adaptive Regression Splines, MARS, Support Vector Machine, SVM, Artificial Neural Networks, ANNs); 2) regression models (Multiple Linear Regression, MLR, MARS, SVM, ANNs, PLS) and 3) a combined model of Orthogonal Partial Least Squares Discriminant Analysis (OPLS-DA). This is the first time I see comparison with all these models and this is what I found original and innovative. Regarding improvements I saw that legibility was not clear in the first 2 figures. I also recommended the reduction of the no. of tables which is too many to read. Conclusions and references are fine.

Reviewer 3 Report

The manuscript presents the use of DSC to identify adulterations in flaxseed oil with rapeseed oil. The paper is inadequate in its present form because some major issues must be considered.

1. The samples are insufficient to calibrate complex models like SVM, ANN, and others. There are only 24 original samples, and with few samples, it is only possible to show the proof-of-concept of the ability of DSC curves to identify adulterations in flaxseed oils. These samples could be employed for PCA and linear regression, but reliable complex models can not be adjusted with few samples.

2. The work did not use data fusion. Data fusion is the simultaneous data analysis from different sources for the same samples, which could be combined in low-, mid-, or high-level approaches. The manuscript deals only with data from DSC, so it is not a data fusion approach.

3. The DA method is not recommended for authentication problems, especially in this case few samples. One-class modeling must be employed; try to use DD-SIMCA or one-class SVM.

Some minor revisions are necessary.

Reviewer 4 Report

This article is well written and fits with journal scope, mainly for detecting the adulteration practice using DSC and chemometrics. Some comments to be addressed are:

1. Please give the short conclusion in the last abstract

2. Gap analysis regarding the importance of DSC over other analytical methods are recommended to be added

3. The novelty of this study should be highlighted in Introduction

4. The method of DSC measurement is missing from reference. If available, please add reference for treacibility.

5. the quality of figure should be improved.

6. The authors should discuss the acceptance criteria during the validation modelling. How about accuracy and precision of validation results.

7. The overfitting of the regression model should be discussed